# Unveiling new disease, pathway, and gene associations via multi-scale neural network

**Thomas Gaudelet**[1], **Noël Malod-Dognin**[2], **Jon Sánchez-Valle**[2], **Vera Pancaldi**[2,3,4], **Alfonso Valencia**[2,5], **Nataša Pržulj**[1,2,5]\*

**1** Department of Computer Science, University College London, London, United Kingdom, **2** Barcelona Supercomputing Center (BSC), Barcelona, Spain, **3** Centre de Recherches en Cancérologie de Toulouse (CRCT), UMR1037 Inserm, ERL5294 CNRS, 31037, Toulouse, France, **4** University Paul Sabatier III, Toulouse, France, **5** ICREA, Pg. Lluis Companys, Barcelona, Spain

\* natasa@bsc.es

## Abstract

Diseases involve complex modifications to the cellular machinery. The gene expression profile of the affected cells contains characteristic patterns linked to a disease. Hence, new biological knowledge about a disease can be extracted from these profiles, improving our ability to diagnose and assess disease risks. This knowledge can be used for drug re-purposing, or by physicians to evaluate a patient's condition and co-morbidity risk. Here, we consider differential gene expressions obtained by microarray technology for patients diagnosed with various diseases. Based on these data and cellular multi-scale organization, we aim at uncovering disease–disease, disease–gene and disease–pathway associations. We propose a neural network with structure based on the multi-scale organization of proteins in a cell into biological pathways. We show that this model is able to correctly predict the diagnosis for the majority of patients. Through the analysis of the trained model, we predict disease–disease, disease–pathway, and disease–gene associations and validate the predictions by comparisons to known interactions and literature search, proposing putative explanations for the predictions.

## Introduction

A disease is often described by symptoms and affected tissues. However, to give a definite diagnosis, physicians often need to analyse patient samples (e.g., blood samples, or biopsies) for characteristic disease indicators, commonly referred to as disease biomarkers. These may include disregulated genes, or pathways [1, 2]. Taking into consideration the history of a patient's past and present conditions identifying genetic predispositions, as well as considering associations between diseases, aid in achieving accurate diagnostics and treatments [3]. By also taking advantage of the increasing availability of large scale molecular data, precision medicine aims at improving the understanding of the molecular base of diseases on individual basis, as well as the relationships between different conditions [4, 5]. The benefits from such work are multiple and include drug re-purposing and identification of new disease biomarkers to improve treatments and diagnoses.

Many studies have investigated disease–gene and disease–pathway associations with the objective of improving diagnoses [6–9]. For instance, Zhao *et al.* [8] propose a ranking of disease genes based on gene expression and protein interactions using Katz-centrality. Hong

**Data Availability Statement:** The unprocessed data is available from GEO and ArrayExpress databases. The GEO and ArrayExpress projects accession numbers, the processed data and all

result data are available online at https://life.bsc.es/
iconbi/MultiScaleNN/index.html.

**Funding:** This work was supported by the
European Research Council (ERC) Consolidator
Grant 770827, UCL Computer Science, the
Slovenian Research Agency project J1-8155, the
Serbian Ministry of Education and Science Project
III44006, the Prostate Project, the Fondation
Toulouse Cancer Sante and Pierre Fabre Research
Institute as part of the Chair of Bio-Informatics in
Oncology of the CRCT, PhD Fellowship (BES-2016-
077403) and the Spanish Ministry of Economics
and Competitiveness (BFU2015-71241-R)

**Competing interests:** The authors have declared
that no competing interests exist.

*et al.* [9] design a tool that identifies significantly disrupted pathways by comparing patient gene expression against controls collected from other experiments. Cogswell *et al.* [10] identify putative gene and pathway biomarkers through change in miRNA in Alzheimer's disease. In specific cancers, Abeel *et al.* [6] use support vector machines and ensemble feature selection methods to select putative gene biomarkers. Ciucci *et al.* [11] developed a general purpose algorithm based on the analysis of PCA loadings that can be used to identify genes that discriminate between conditions from expression data.

A key issue is that most of these studies consider diseases in isolation, i.e. comparing patients having a disease of interest to healthy individuals while the predicted biomarkers could be shared between various diseases. This limits the discriminative potential of such studies for accurate diagnoses. Indeed, network medicine has shown that diseases can share significant molecular background, as evidenced by numerous studies based on patient historical records [3, 12, 13], biological knowledge of the diseases [4, 5, 14], or patient gene expression profiles [15]. For instance, Goh *et al.* [4] build a disease network, which connects diseases that share at least one gene which when mutated is linked to both conditions. Lee *et al.* [5] construct a disease network of metabolic diseases, connecting pairs of diseases for which associated mutated enzymes catalyze adjacent metabolic reactions. Hidalgo *et al.* [12] take a different approach by building a disease network based on disease co-morbidities, i.e. two diseases are connected if they tend to co-occur significantly in the patient populations. They used Medicare records of elderly patients to build the network. He *et al.* [14] propose PCID (Predicting Comorbidity by Integrating Data), an approach to predict disease co-morbidities by aggregating disease similarity scores derived from different data including protein–protein interactions (PPIs), pathways, and functional annotations.

Sánchez-Valle *et al.* [15] define a disease network, named the Disease Molecular Similarity Network (DMSN) based on patient's differential expression profiles. In their study, the DMSN is generated using positive and negative relative molecular similarities (RR) to measure disease similarity and dissimilarity, respectively, that is then interpreted as an estimate of risk. First, a patient-patient similarity network is generated based on the similarities of patient's differential expression profiles. Next, using the relative similarity score, diseases are related to each other. The resulting network is directed and each edge is associated to a positive or negative label indicating either an increased or decreased risk of developing the target disease if the patient has the source disease. The underlying assumption is that having a given disease can increase the risk of developing a disease characterized by a similar gene expression profile.

In these various approaches, a key issue is that either a single data source is used, such as disease–gene mutational data [4], or no new biological knowledge about a specific disease could be derived from the results (e.g., PCID [14]).

In this work, we propose an integrative framework based on artificial neural networks (NN) to predict disease–disease links, as well as disease–pathway and disease–gene associations. We train the model to predict patients' diagnoses based on differential gene expression. The NN's structure is designed to mimic the cellular multi-scale functional organization by integrating gene–pathway annotations. This approach follows on from a body of work aiming to build neural networks based on prior information defining the structure of the network [16]. For instance, Ma *et al.* [17] build a neural network using the Gene Ontology [18] directed acyclic graph as a template for the connections. The neural network, named DCell, is then trained to predict phenotype related to cellular fitness from genotype data. The trained DCell predicts cellular growth almost as accurately as laboratory observations.

We show that our framework achieves good predicting performances on our dataset. By analysing the trained NN, i.e. the underlying weight matrices, we show that we can extract biological knowledge relevant for each disease. Specifically, we use the trained NN to predict

novel disease–pathway and disease–gene links and from those predictions we extract disease similarity score used to identify putative co-morbidities. We show that our predictions are biologically relevant against established ground-truths and verify the top predictions through manual literature curation ensuring that the sources do not use the same data, to mitigate the risk of argument circularity.

# 1 Material & methods

## 1.1 Datasets

We use the original dataset of patient gene expressions used by [15] in which each patient is associated to a single disease (see S1 File for details). For each patient, we define a differential gene expression vector of size corresponding to the number of genes and in which the $i^{th}$ entry is equal to 1, −1, or 0 depending on whether the $i^{th}$ gene is significantly (with 5% cut-off) over-, under-, or normally expressed, respectively, for that patient (see S1 File for details).

Pathway annotations were collected from Reactome database [19] (accessed December 2018). Only the lowest pathways in the hierarchy are considered to avoid dealing with pathway interactions (i.e. pathways containing other pathways). Of those pathways, only the ones that have a Traceable Author Statement (TAS) are kept. In total, we consider 1, 708 pathway annotations.

The final dataset contains 4, 788 samples (patients) diagnosed by one of 83 diseases (see S1 Table in S1 File for cohort distribution). In total, 20, 525 genes have their expressions measured, but only a subset is used as input to our method described in the following section, as we restrict ourselves to genes associated to at least one pathway, which leaves 9, 247 genes.

## 1.2 Neural network based data–integration framework

We propose a neural network (NN) predicting a patient diagnosis based on differential gene expression. The structure of the neural network is based on molecular organization, more specifically gene–pathway annotations downloaded from Reactome (see Fig 1). We integrate molecular organization data into our model to reflect the idea that complex diseases, such as cancer, can be the results of the perturbations of groups of genes, as opposed to a single gene. Using Reactome data allows us to incorporate prior knowledge into our model in the form of biologically meaningful groupings of genes.

A neural network can be expressed as a series of matrix multiplications interleaved with non-linear function (See S1 File for more details). Here, we use the softmax function [20] as the last non-linear function of the NN (common choice for multiclass classification problems) and the hyperbolic tangent non-linear function for hidden layers to allow a hidden unit to have values lying in [−1, 1] to represent up- and down-regulations. We use the classical cross-entropy function to define the loss function. Our neural network architecture has only one hidden layer capturing gene–pathway links. Hereafter, we refer to this model as GPD (for Gene–Pathway–Disease).

As reference model, we use a multiclass logistic regression (MLR; no hidden layer). MLR and GPD have a different number of parameters (or free weights): (573, 314) for MLR and (137, 838) for GPD. Note that, due to this imbalance, we do not expect GPD to outperform MLR in the diagnosis prediction task.

To train both GPD and MLR, we first perform a 10-fold cross-validation to fix the number of training epochs. To fix the number of training epochs, we compute the average number of epochs at which the test loss is the smallest across the runs (see S1 Fig and S2 Fig in S1 File). As the dataset is imbalanced, we use stratification to split the data, ensuring that at least one patient per disease is in the test set. Using this number of epochs, we perform another 10-fold

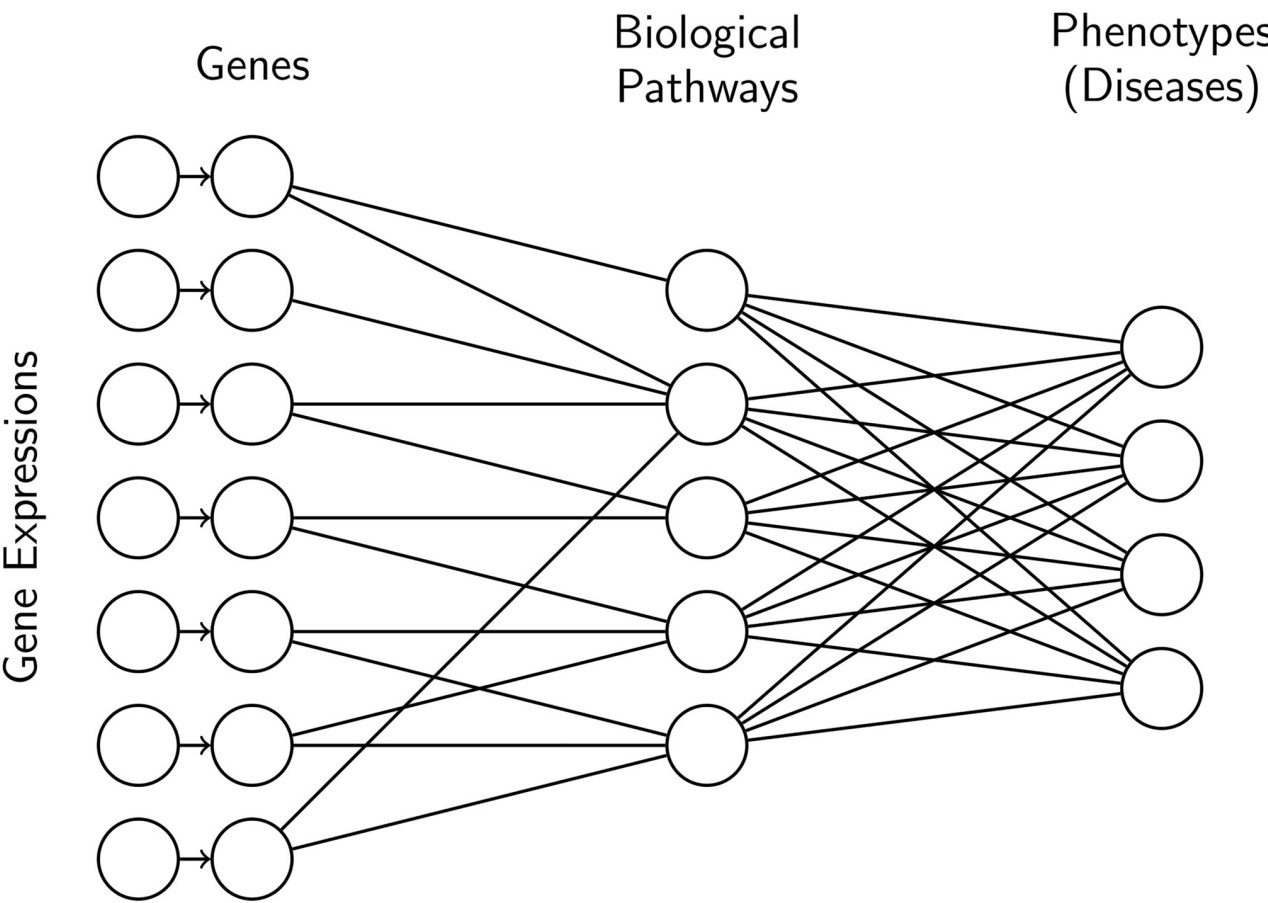

Genes

Biological Pathways

Phenotypes (Diseases)

Gene Expressions

**Fig 1. Example of neural network architecture.** For the first layer, the connections are defined by biological information, i.e. a unit representing a gene is connected to all the biological pathways that the gene is involved in. We do not add any prior knowledge on the last layer, thus it is fully connected.

cross-validation to evaluate the performance of our models. We use the Adam optimizer [21] with learning rate 0.01 and the layer weights are initialized to small values using the initialization process proposed by He *et al.* [22]. We investigated the addition of classical regularisation techniques—L1, L2, and dropout regularisation—as a mean to reduce the capacity of the model to overfit. We found that the performances were better without any regularisations (see S2 Table in S1 File).

The neural networks are implemented with Tensorflow [23].

## 1.3 Predicting disease–disease, disease–pathway, and disease–gene relationships

To identify associations between diseases and genes or pathways, we perform sensitivity analysis [24] (see S1 File). The association score between disease *d* and unit *u* (representing a gene, or a pathway) is measured by the intensity of the local variation of the output unit associated with *d* with respect to perturbation of *u*. Intuitively, this score measures how the prediction score of disease *d* is affected by disregulation of a gene/pathway: we quantify the change in one disease score induced by a disregulation in the gene expression, or pathway activation. We test this scoring approach for the prediction of disease–gene and disease–pathway associations. In particular, we rank disease–gene and disease–pathway pairs based on this score and test if the

score correlates with known associations through a Precision-Recall and Receiver Operating Characteristic (ROC) analysis. Score thresholds could be inferred from the precision-recall curves. These thresholds would be uninformative in general since they are tied to specific runs and datasets. Hence, we focus on manual validation of the top scoring associations.

Based on this score, we represent each disease by a set formed by the $k_{genes}$ highest scoring genes and a set containing the $k_{pathways}$ highest scoring pathways. We then score disease–disease associations using the Jaccard Index of their sets. The Jaccard Index of two sets $S_1$ and $S_2$ is defined as $\frac{|S_1 \cap S_2|}{|S_1 \cup S_2|}$, where $|\cdot|$ represents the cardinality of a set. Following on from similar approach used in DisGeNET [25], we interpret those associations as co–morbidities. The number of highest scoring pathways and genes considered is set to 150 and 300, respectively, as those numbers gave the best results.

## 2 Results & discussion

### 2.1 Classification performances

To validate the relevance of our model, we verify that the classification performances are at least on par with competing methods: MLR, Random Forest (RF), Bernoulli Naive Bayes (nB), and Support Vector Machine (SVM) algorithms (we use the implementation available through the scikit-learning python package [26]). We perform 10-fold cross-validation for the algorithms to fix the hyperparameters (numbers of trees 100, smoothing parameter 0.001 and penalty parameter 100, respectively) and retain the best performing models in terms of cross-entropy loss (the objective function of the neural networks).

We evaluate performances by computing 3 different scores: cross-entropy loss (CEL), micro- and macro-average precision ($Pre_\mu$ and $Pre_M$). $Pre_\mu$ give a measure of the overall precision of each classifier, while $Pre_M$ gives an average of the precisions across the different classes (diseases). The details of each score can be found in S1 File.

We observe that the neural networks; MLR and GPD, give better, or at least on-par, performances when compared to RF, nB, and SVM classifiers as measured by our three metrics (see Table 1). This observation justifies the relevance of our GPD model. The best model appears to be the multinomial logistic regression (MLR), which corresponds to the most complex neural network model in terms of the number of parameters (or degree of freedom), since MLR has $\sim 4$ times more parameters than GPD. This analysis shows that using biological knowledge to guide the structure of neural networks, in the limit of the models proposed, does not improve classification performance compared to the multiclass logistic regression (MLR) and only offers slight improvement when compared to a RF classifier (see Table 1). However, we show, in the following Sections, that the trained GPD models can be more successfully exploited than MLR to extract biological information. Note as well that the gene–pathway information on

**Table 1. Performances of different classifiers in terms of cross-entropy loss (CEL), micro- and macro-average precisions ($Pre_\mu$ and $Pre_M$, respectively).** Each score is computed across the 10-fold cross-validation and we provide the standard deviation. Bold scores highlight the best scores for each metric.

| Algorithm | CEL | $Pre_\mu$ | $Pre_M$ |
|-----------|-----|-----------|---------|
| GPD | 1.09±0.06 | 0.80±0.01 | 0.71±0.02 |
| MLR | **1.01 ± 0.07** | **0.84 ± 0.01** | **0.76 ± 0.01** |
| RF | 1.56±0.24 | 0.80±0.01 | 0.70±0.03 |
| nB | 10.63±0.55 | 0.66±0.01 | 0.60±0.02 |
| SVM | 1.42±0.04 | 0.72±0.02 | 0.59±0.02 |

which GPD relies is both noisy and incomplete, as biological data often is, and that performances should improve as knowledge improves.

Hereafter, we consider for each model (GPD and MLR) the trained NN that gave the lowest cross-entropy loss during the 10-fold cross validation.

## 2.2 Our GPD model uncovers molecular mechanisms of diseases

To uncover molecular mechanisms of disease, i.e., genes and pathways that are associated to specific diseases, we extract predictions from MLR and GPD using the approach described in Section 1.3. We test the performance of our disease–pathway and disease–gene associations predictions by comparing against established databases. We investigate the top predictions of the GPD model through manual search of the literature.

**Predicting disease–gene associations.** For each model, we compute disease–gene association scores as described in Section 1.3 and we test the validity of our predictions against DisGeNET database [25]. We compare the entire set of predictions against two baselines: the Frequency of Differential Expression (FDE) and the approach introduced by Zhao *et al.* [8] for *de novo* disease–gene association prediction (Katz). Those methods are detailed in the S1 File.

We use precision–recall and ROC curves to evaluate the performance of our approach and compute the areas under the curves (see Table 2 and S3 Fig and S4 Fig in S1 File). Interestingly, we observe that the FDE score is a poor predictor of disease–gene associations. We further observe that GPD is the best performing models for this task with Katz coming second. The relatively poor overall performances can be partially attributed to the incompleteness of the reported disease–gene associations in DisGeNET. To corroborate this hypothesis, we search the literature for support for the top 10 predicted disease–gene associations by the best performing model, GPD (see Table 5). Note that none of those associations are reported in DisGeNET.

We are able to find literature support for 70% of the top 10 predicted disease–gene associations (see Table 3). Furthermore, we find indications that some of our top-scoring, non-validated predictions could be relevant, such as the associations of asthma with UBB and amyotrophic lateral sclerosis (ALS) with PSMD13. Ubiquitin B (UBB) belongs to the ubiquitin-proteasome (UPS) and it is known that aberration in the UPS is responsible for inflammatory and autoimmune diseases such as asthma [27]. Moreover, ALS onsets occur typically after age 50 and manifest partially through muscle weakness. PSMD13 is linked to aging [28] and high expression of the gene has been found in skeletal muscle of athletes [29], suggesting that under-expression could be a sign of muscle weakness.

These results validate the relevance of our framework for de novo disease–gene association prediction and confirm the incompleteness of DisGeNET.

**Predicting disease–pathway associations.** For our GPD model, we compute disease–pathway association scores as described in Section 1.3 and we test the validity of our predictions by comparison with CTD database [30]. As a baseline, we consider disease–pathway

Table 2. Performance in terms of area under the ROC (AUROC) and area under the precision–recall (AUPRE) for the prediction of disease–gene associations for each methods.

|  | AUROC | AUPRE |
|---|---|---|
| GPD | **0.59** | **$8.5e^{-3}$** |
| MLR | 0.52 | $6.3e^{-3}$ |
| Katz | 0.55 | $7.3e^{-3}$ |
| FDE | 0.50 | $5.3e^{-3}$ |

**Table 3. Top 10 disease–gene predicted by GPD.**

| Disease | Gene | Literature support |
|---|---|---|
| Asthma | UBB | |
| Schizophrenia | RHOA | PMID:16402129 |
| Alzheimer's disease | FGF23 | PMID:26674092 |
| Autistic disorder | FGF20 | PMID:19204725 |
| Prostate cancer | RPS27A | PMID:15647830 |
| Amyotrophic lateral sclerosis | PSMD13 | |
| Amyotrophic lateral sclerosis | CASP3 | PMID:11715057 |
| Chronic obstructive pulmonary disease | SKP1 | PMID:23713962 |
| Autistic disorder | PSMB2 | |
| Irritable bowel syndrome | PSMA1 | PMID:28717845 |

scores corresponding to the average FDE (AFDE) of genes within the pathway for patients having the disease.

We evaluate the results as done previously for disease–gene associations (see Table 4 and S5 Fig and S6 Fig in S1 File). We observe that GPD convincingly outperforms AFDE. The seemingly poor performances of both approaches can partially be attributed to the incompleteness of CTD database. To test this hypothesis, we search the literature for support for the top 10 disease–pathway associations predicted with our GPD (see Table 5). Note that none of these associations predicted are reported in CTD database.

We find literature support for 7 out of the top 10 predicted disease–pathway associations (see Table 5). Furthermore, we find indications that some of our top-scoring, non-validated predictions could be relevant, such as the association of autistic disorder with the lactose synthesis pathway (R-HSA-5653890) and the association of Schizophrenia with pathway R-HSA-5683371 linked to microphtalmia. The lactose synthesis pathway (R-HSA-5653890) contains three genes: LALBA, SLC2A1, and B4GALT1. All of those genes might be associated with autistic disorders. One patented method to detect autistic disorder (US20140349977A1) includes LALBA as one of the genes of interest. SLC2A1 mutation has been reported in patients diagnosed with autism [31]. And finally, B4GALT1 has been linked with developmental disorders [32]. The pathway R-HSA-5683371 is linked to the eye disease microphtalmia. It is known that schizophrenia is linked to eye abnormalities [33]. Among the 28 genes involved in that pathway, 12 have been linked to the disease in the literature (GOT2, PDHA1, DLD, GCSH, DLAT, PDHB, DAO, OGDH, DHTKD1, GNMT, DDO, PRODH2).

These results show the relevance of our framework for de novo disease–pathway associations prediction despite relatively low retrieval scores against the ground–truth.

## 2.3 Our GPD model predicts disease–disease relationships

We rank disease–disease pairs based on the score described in Section 1.3 and test our results against a high confidence co-morbidity disease network obtained from a large cohort study by

**Table 4. Performance in terms of area under the ROC (AUROC) and area under the precision–recall (AUPRE) for the prediction of disease–pathway associations for each methods.**

| | AUROC | AUPRE |
|---|---|---|
| GPD | **0.53** | **$8e^{-2}$** |
| AFDE | 0.47 | $6e^{-2}$ |

**Table 5. Top 10 disease–pathway predictions derived from GPD.**

| Disease | Pathway R-HSA- | Literature support |
|---|---|---|
| Autistic disorder | 5653890 | |
| Irritable bowel syndrome | 532668 | PMID:20338921 |
| Irritable bowel syndrome | 391906 | PMID:16835707 |
| Type 2 diabetes mellitus | 499943 | doi:10.2337/diabetes.51.2007.S363 |
| Asthma | 391906 | PMID:8603274 |
| Schizophrenia | 71288 | PMID:22465051 |
| Major depressive disorder | 8934903 | PMID:27063986 |
| Type 2 diabetes mellitus | 8939245 | PMID:19667185 |
| Schizophrenia | 5683371 | |
| Sjogren's syndrome | 389661 | |

Hidalgo *et al.* [12]. We compare our method against DMSN network [15], restricted to our set of diseases, and three alternatives baselines. For the first alternative, we compute disease–disease association score using our approach defined in Section 1.3 on the trained MLR network, representing each disease by the top 300 highest scoring genes (which gave the best results based on grid search). The final two baselines associate to each disease–disease pair a Jaccard Index score based on 1) the set of genes associated to each disease in DisGeNET [25] and 2) the set of pathways associated to each disease in CTD database [30]. The results of the comparison are presented using a precision–recall curve (see Fig 2).

We observe that our approach outperform convincingly the other approaches in the task of retrieving existing co-morbidity links between diseases with over 10% increase compared to DMSN and 30% improvement over DisGeNET in terms of area under the precision–recall curve (auprc). These results strongly support our methodology. The scoring based on disease–gene is performing better than disease–pathway, hence we investigate the top 10 scoring disease–disease associations derived from it (see Table 6).

We present and discuss below literature support for the predicted associations between the diseases.

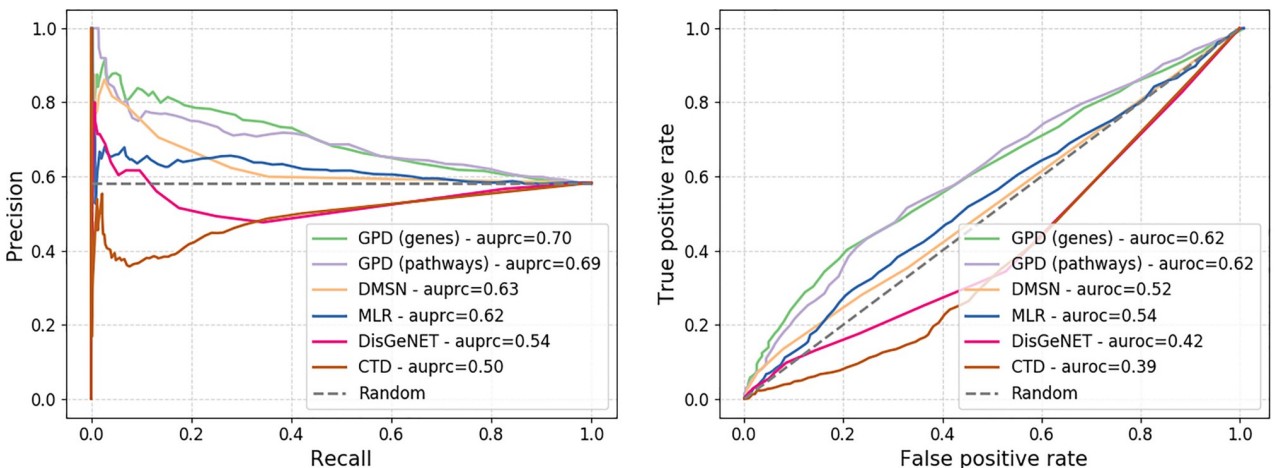

**Fig 2. Precision–recall (top) and ROC (bottom) curves of the test against the disease co-morbidity network built by Hidalgo *et al.* [12].**

**Table 6. Top 10 disease–disease links predicted using our approach based on the trained GPD.**

| Disease 1 | Disease 2 |
|---|---|
| Atrial fibrillation | Vitiligo |
| Atrial fibrillation | Peripheral vascular disease |
| Alcoholic hepatitis | Osteosarcoma |
| Rhabdoid cancer | Medulloblastoma |
| Cornelia de Lange syndrome | Vitiligo |
| Peripheral vascular disease | Vitiligo |
| Atrial fibrillation | Osteosarcoma |
| Leishmaniasis | Alcoholic hepatitis |
| Sotos syndrome | Vitiligo |
| Follicular lymphoma | Osteosarcoma |

Atrial fibrillation has been linked in the literature to thyroid disease [34] which is known to be co-morbid with vitiligo [35]. Atrial fibrillation and peripheral vascular disease are well known co-morbid conditions [36]. Alcoholism has been linked to the onset of some cancers notably implicating the transcription factor Nanog which itself has been linked to osteosarcoma [37]. Additionally, a drug used to treat alcoholism, Disulfiram, has recently been proposed as a potential treatment for osteosarcoma [38]. Those information put together suggest shared molecular background for the two conditions. Rhabdoid cancer is a rare form of aggressive cancer affecting young children and with very poor prognostic, which makes it difficult to evaluate co-morbid conditions. However, rhabdoid cancer is frequently mistaken for medulloblastoma indicating some similarity [39]. We found no evidence in the literature supporting a connection between the rare Cornelia de Lange syndrome and Vitiligo. Some studies have observed significant co-morbidity between vitiligo and psoriasis and the combination of the two has been linked to cardiovascular diseases, which include peripheral vascular disease [35, 40]. Atrial fibrillation and cardiac complications have been observed as the result of osteosarcoma [41, 42]. Leishmaniasis and alcoholic hepatitis are an unlikely co-morbid connection since it would require a patient both to have been infected by parasites of the Leishmania type and have had excessive alcohol intake. However, both disease affects the liver and leishmaniasis has sometimes been misdiagnosed for cirrhosis [43] which suggests that the two diseases might share some similar molecular processes that we would be capturing here. A case of co-occurence of Sotos syndrome and vitiligo has been reported in the medical literature [44]. It has been postulated that non-Hodgkin's lymphoma (which include follicular lymphoma) and osteosarcoma share underlying mechanisms [45]. Additionally, miR-202 has been identified as a potential tumor suppressor for both conditions [46].

Through this analysis, we have shown that most predicted pairs have either been observed co-occurring or can be connected through underlying mechanisms, thus validating our approach.

## 3 Conclusions

In this study, we propose a multi-scale neural-network based framework that integrates gene expression data associated to diseases with gene–pathway information. Our integrative framework allows for simultaneously uncovering novel disease-disease associations and disease molecular mechanisms from patient gene expression profiles through the analysis of trained neural networks. We show that GPD achieve good diagnosis prediction on our dataset showing the validity of our integrative process. Furthermore, we show that the associations

predicted from the trained models are biologically meaningful and supported by the literature, thus validating our approach.

While our disease molecular mechanisms are supported by the current knowledge about these diseases, a next step would be to identify among the predicted genes and pathways suitable biomarkers and drug targets that could be used to improve diagnosis, prognosis, and treatment. We leave this for future work. Also, while our multi-scale NN framework integrates the hierarchical functional organization of a cell (from genes to biological pathways), our methodology can be extended to include any dataset pertaining to diseases of interest, e.g., uncovering molecular mechanisms of cancer from patient somatic mutation profiles or linking diseases to non-coding RNA.

Finally, while we focus on patient data with application to diseases, our methodology can be extended to integrate additional omics data with the objective of getting more biologically accurate models for the analyses of patients, tissues, and cells. Some further applications include studies of diseases linked to a specific tissue, studies of cell's specialization, and any study that can benefit from the integration of the hierarchical functional organization of cells.

## Supporting information

**S1 File. Supplementary methods, figures, and tables. Project webpage** https://life.bsc.es/iconbi/MultiScaleNN/index.html.
(PDF)

## Author Contributions

**Conceptualization:** Thomas Gaudelet.

**Data curation:** Thomas Gaudelet, Jon Sánchez-Valle, Vera Pancaldi.

**Formal analysis:** Thomas Gaudelet.

**Funding acquisition:** Alfonso Valencia, Nataša Pržulj.

**Investigation:** Thomas Gaudelet.

**Methodology:** Thomas Gaudelet, Noël Malod-Dognin.

**Project administration:** Nataša Pržulj.

**Software:** Thomas Gaudelet.

**Supervision:** Noël Malod-Dognin, Alfonso Valencia, Nataša Pržulj.

**Validation:** Thomas Gaudelet.

**Visualization:** Thomas Gaudelet.

**Writing – original draft:** Thomas Gaudelet.

**Writing – review & editing:** Noël Malod-Dognin, Jon Sánchez-Valle, Vera Pancaldi, Alfonso Valencia, Nataša Pržulj.

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
