## [Decision Letter · Decision Letter 0]

21 Jan 2020

PONE-D-19-28184

Unveiling new disease, pathway, and gene associations via multi-scale neural networks

PLOS ONE

Dear Mr Gaudelet,

Thank you for submitting your manuscript to PLOS ONE. After careful consideration, we feel that it has merit but does not fully meet PLOS ONE’s publication criteria as it currently stands. Therefore, we invite you to submit a revised version of the manuscript that addresses the points raised during the review process.

We would appreciate receiving your revised manuscript by Mar 06 2020 11:59PM. To enhance the reproducibility of your results, we recommend that if applicable you deposit your laboratory protocols in protocols.io, where a protocol can be assigned its own identifier (DOI) such that it can be cited independently in the future. For instructions see: http://journals.plos.org/plosone/s/submission-guidelines#loc-laboratory-protocols

We look forward to receiving your revised manuscript.

Kind regards,

Carlo Vittorio Cannistraci

Academic Editor

PLOS ONE

Additional Editor Comments (if provided):

Dear Authors

Congratulations

You article requires only a minor revision.

Best

CVC

Journal Requirements:

2. In your Data Availability statement you have specifed:

"The processed data and all result data are available online at " ext-link-type="uri" xlink:type="simple">https://life.bsc.es/iconbi/MultiScaleNN".

But this link can not be found. PLOS defines a study's minimal data set as the underlying data used to reach the conclusions drawn in the manuscript and any additional data required to replicate the reported study findings in their entirety. All PLOS journals require that the minimal data set be made fully available. For more information about our data policy, please see http://journals.plos.org/plosone/s/data-availability.

Reviewers' comments:

Reviewer's Responses to Questions

**Comments to the Author**

1. Is the manuscript technically sound, and do the data support the conclusions?

Reviewer #1: Yes

Reviewer #2: Yes

2. Has the statistical analysis been performed appropriately and rigorously? 

Reviewer #1: Yes

Reviewer #2: N/A

3. Have the authors made all data underlying the findings in their manuscript fully available?

Reviewer #1: Yes

Reviewer #2: Yes

4. Is the manuscript presented in an intelligible fashion and written in standard English?

Reviewer #1: Yes

Reviewer #2: Yes

5. Review Comments to the Author

Reviewer #1: Really interesting work from the authors, which trained a neural network-based method with prior biological knowledge and whose architecture is based on molecular organization with one hidden layer representing gene-pathway links.

The purpose of the NN model is to unveil disease-gene, disease-pathway and disease-disease associations. Therefore, the authors termed it GPD (for Gene, Pathway and Disease). From a selected dataset of patient-disease interaction (one patient had one disease), GPD was compared to other NN model based on multinomial logistic regression (MLR), random forest, naïve bayes and support vector machine classifiers in the task for patient-disease classification. The MLR model outperformed the rest of the methods followed by GPD. The authors state that this might be due to two main factors: the degree of freedom of MLR which is ~4 times higher than GPD, and that the biological information from where GPD was trained is noisy and incomplete. Nonetheless, valuable biological information can be extracted from the trained GPD model, where literature-validated associations between disease-gene, disease-pathway and disease-disease were found.

Minor comments:

1.In the introduction section, line 20: “In specific cancers, Abeel et al. [6] use support vector machines and ensemble feature selection methods to select putative gene biomarkers.”

You might consider also to refer to articles that show methods that can work on different omic signatures for cancer in general like Ciucci et al., 2017 Scientific reports.

2.The authors gave an association score for the prediction of disease-disease, disease-pathway and disease-gene relationships, which was lately use to obtain the highest 10 predictions under the three mentioned relationships, validating the predictions by literature search. It would be interesting to known the extent to which this score is “high-enough” to obtain valuable information from the trained GPD model, since this score magnitude is not mentioned on the manuscript.

Reviewer #2: === General Comments

The authors present a neural network incorporating pathway information for disease classification based on differentially expressed genes. The core of the work is less the correct prediction than the interpretation of the network's choice and thus enabling the deciphering of disease-disease networks and a shift in the causal direction. The paper gives a comprehensive introduction into existing work, is well written and scientifically sound. The developed method is compared against a Multi-label logistic regression baseline model. The model itself was trained following common standards with early stopping as regularization based on a cross-validation.

=== Minor Comments

* The authors should clarify their method for the prediction of disease-(disease/pathway/gene) relationships. What is meant with the intensity of local variation of d with respect to perturbation of u? Are there thresholds involved, e.g., when is the intensity high enough to create the relationship.

* Regularization: The authors utilize early stopping, but no other regularization. Here, particular considering the number of genes vs. the number of samples might pose a problem. Have the authors considered heavy drop-out regularization of the input layer, which might boost the necessity to use the pathway layer since it cannot any longer rely on the presence of single genes?

* Connected to the point above, the authors should depict the learning behavior of their network wrt. the loss training vs. validation. Is the observed behavior expected, or does it indicate over- or underfitting.

6. PLOS authors have the option to publish the peer review history of their article (what does this mean?). If published, this will include your full peer review and any attached files.

Reviewer #1: No

Reviewer #2: No

---

## [Author Response · Author response to Decision Letter 0]

13 Feb 2020

Dear Editor, 

Thank you for inviting us to submit a revised version of our manuscript titled “Unveiling new disease, pathway, and gene associations via multi-scale neural network”. 

We revised our paper to address the reviewers’ suggestions. Our responses to the reviewers’ comments are summarized below. 

We believe that the manuscript is now suitable for publication in PloS One, and look forward to hearing from you. 

Sincerely yours, 

Nataša Pržulj 

Reviewer 1:

1. In the introduction section, line 20: “In specific cancers, Abeel et al. [6] use support vector machines and ensemble feature selection methods to select putative gene biomarkers.”

You might consider also to refer to articles that show methods that can work on different omic signatures for cancer in general like Ciucci et al., 2017 Scientific reports.

Response: We followed the suggestion of the reviewer and in our revised manuscript we updated the introduction accordingly (page 2, paragraph 1).

2. The authors gave an association score for the prediction of disease-disease, disease-pathway and disease-gene relationships, which was lately use to obtain the highest 10 predictions under the three mentioned relationships, validating the predictions by literature search. It would be interesting to known the extent to which this score is “high-enough” to obtain valuable information from the trained GPD model, since this score magnitude is not mentioned on the manuscript.

Response: We thank the reviewer for the comment. There is no out-of-the-box threshold that can answer this question and we chose here to validate our scoring strategy with manual literature curation and precision-recall analysis against an independent ground-truth. However, one could use the precision-recall analysis we ran as a way to estimate how many of the top scoring associations can be considered relevant. We adjusted the main to reflect this. (page 4, paragraph 5)

Reviewer 2: 

1. The authors should clarify their method for the prediction of disease-(disease/pathway/gene) relationships. What is meant with the intensity of local variation of d with respect to perturbation of u? Are there thresholds involved, e.g., when is the intensity high enough to create the relationship.

Response: We thank the reviewer for the comment. The score used to identify molecular links to the disease rests on the analysis of variations of the output of a function (e.g., a neural network or part of one) with respect to the variation of one of its input (e.g., gene expression). Intuitively, we assume that the trained neural network links a gene, for instance, to a disease if a change in the expression of the gene leads to a variation in the disease score given by the output of the neural network. Effectively, this is the analytical equivalent of manually varying the expression of each gene and measuring the impact on the score and prediction of each disease. We postulate here that the higher the induced variation, the more likely there is an actual biological connection between the gene and disease. To assess this, we rank disease-gene pairs based on this score and we evaluate how well this ranking captures known interactions with precision-recall and receiver operating characteristic analysis as well as manual literature curation. We updated the manuscript to clarify those points. (page 4, paragraph 5)

The second part of the comment relating to threshold is the same as the second comment of the first reviewer and we refer the reader to the response above.

2. Regularization: The authors utilize early stopping, but no other regularization. Here, particular considering the number of genes vs. the number of samples might pose a problem. Have the authors considered heavy drop-out regularization of the input layer, which might boost the necessity to use the pathway layer since it cannot any longer rely on the presence of single genes?

Response: We thank the reviewer for the suggestion. We had investigated the addition of L2 and L1 regularisations with a cross-validation but observed that the models without were performing better in terms of cross-entropy loss thus we removed any. Heavy dropout on the input layer was not investigated, following the reviewer suggestion, we have run test with it and observed that similarly it was not leading to improvements in terms of cross-entropy loss or overfitting. We updated the text to mention the tests with addition of L1, L2, or dropout regularisations and added a supplementary table with cross-validation scores for each regularisation. (page 4, paragraph 3 and Supplementary Table 2)

3. Connected to the point above, the authors should depict the learning behavior of their network wrt. the loss training vs. validation. Is the observed behavior expected, or does it indicate over- or underfitting.

Response: We followed the reviewer suggestion, and in our revised manuscript we show the learning behaviour of the neural networks (Supplementary Figures 1 and 2). Indeed, the plots suggest some overfitting, however, as mentioned above, the performances remain better than when regularisation is added to address the overfitting.

---

## [Editor Report · Decision Letter 1]

16 Mar 2020

Unveiling new disease, pathway, and gene associations via multi-scale neural networks

PONE-D-19-28184R1

Dear Authors

We are pleased to inform you that your manuscript has been judged scientifically suitable for publication and will be formally accepted for publication once it complies with all outstanding technical requirements.

With kind regards,

Carlo Vittorio Cannistraci

Academic Editor

PLOS ONE

Additional Editor Comments (optional):

The Authors addressed all minor concerns of the reviewers and the article is accepted.
---

## [Editor Report · Acceptance letter]

17 Mar 2020

PONE-D-19-28184R1 

Unveiling new disease, pathway, and gene associations via multi-scale neural network 

Dear Dr. Gaudelet:

I am pleased to inform you that your manuscript has been deemed suitable for publication in PLOS ONE. Congratulations! Your manuscript is now with our production department. 

With kind regards,

on behalf of

Dr. Carlo Vittorio Cannistraci 

Academic Editor

PLOS ONE